# Reciprocal and Differential Influences of Mediterranean Diet and Physical Activity on Adiposity in a Cohort of Young and Older than 40 Years Adults

**DOI:** 10.3390/nu16111777

**Published:** 2024-06-05

**Authors:** Andrea Higuera-Gómez, Begoña de Cuevillas, Rosa Ribot-Rodríguez, Rodrigo San-Cristobal, Víctor de la O, Karina Dos Santos, Amanda Cuevas-Sierra, J. Alfredo Martínez

**Affiliations:** 1Precision Nutrition and Cardiometabolic Health, IMDEA-Food Institute, Campus of International Excellence (CEI) UAM+CSIC, 28049 Madrid, Spain; andrea.higuera@alimentacion.imdea.org (A.H.-G.); rosa.rib.rod@gmail.com (R.R.-R.); victor.delao@alimentacion.imdea.org (V.d.l.O.); amanda.cuevas@alimentacion.imdea.org (A.C.-S.); jalfredo.martinez@imdea.org (J.A.M.); 2Centre Nutrition, Santé et Société (NUTRISS), Institut sur la Nutrition et les Aliments Fonctionnels, l’Université Laval (INAF), Québec, QC G1V 0A6, Canada; rodrigo.san-cristobal-blanco.1@ulaval.ca; 3School of Nutrition, Université Laval, Quebec, QC G1V 0A6, Canada; 4Faculty of Health Sciences, International University of La Rioja (UNIR), 26006 Logroño, Spain; 5Josué de Castro Institute of Nutrition, Federal University of Rio de Janeiro, 373, Carlos Chagas Filho Ave, University City, Rio de Janeiro 21941-590, RJ, Brazil; santos.karina@unirio.br; 6Public Health Nutrition Department, Federal University of the State of Rio de Janeiro, 296, Pasteur Ave, Rio de Janeiro 22290-240, RJ, Brazil; 7Biomedical Research Centre for Obesity Physiopathology and Nutrition Network (CIBEROBN), Instituto de Salud Carlos III (ISCIII), 28029 Madrid, Spain

**Keywords:** Mediterranean diet, physical activity, precision nutrition, body mass index, online, adults, healthy aging

## Abstract

Translational research has documented the conjoint beneficial relationships between dietary and physical activity habits concerning weight maintenance. However, the precise interplay between diet and exercise impacting body composition remains unclear, challenging personalized interventions. This study aimed to explore potential interactions and effect modifications of these factors affecting the body mass index (BMI) within an online adult cohort. Data from 11,883 NUTRiMDEA cohort participants were analyzed in this cross-sectional study, categorizing individuals by age, sex, and BMI using linear regression models to assess the interactions between lifestyle factors and adiposity. Significant differences emerged in anthropometry, lifestyle, and health-related quality of life (HRQoL) across categories. The combined effect of diet and physical activity had a greater impact on BMI than physical activity or Mediterranean diet adherence alone, with lower BMI as physical activity levels increased (*β*: −0.5) and adherence to the Mediterranean diet decreased, where a modification effect between them was identified (*β*: −0.28). Participants with lower Mediterranean diet adherence displayed superior BMI when physical activity was low, but when activity levels were higher, their BMI aligned with those with healthier dietary habits. An interaction link between lifestyle factors and BMI was found, showing the differential effects of the Mediterranean diet and physical activity combination concerning adiposity.

## 1. Introduction

Advancements in medicine and nutrition have led to longer lifespans for older adults. However, greater longevity does not guarantee a higher quality of life and healthy aging [1]. In this context, healthy aging has been defined as “the process of developing and maintaining the functional ability that enables well-being in older age” [2]. Multiple determinants play a role in aging, ultimately shaping the likelihood of developing age-related chronic conditions in later years [1]. These factors include genetic predisposition alongside diverse behavioral, environmental, and lifestyle determinants, including dietary and physical activity influences, which have been demonstrated to impact specific biological pathways influencing aging processes and longevity [3].

Obesity and adiposity in the general population is measured as a body mass index (BMI) ≥ 30 kg/m^2^, where excessive body weight has become a worldwide major public health concern associated with diverse burdens in many regions, countries, and ages [4]. This trend poses significant challenges for healthy aging, as overweight and obesity are linked to an increased risk of developing a number of chronic diseases and disabilities [4], including a worse health-related quality of life (HRQoL) as a global health marker [5].

Lifestyle could be considered the most relevant and modifiable factor with an impact on health [6,7]. Scientific evidence supporting the correlation between diet and health is widely acknowledged [8]. Multiple studies have indicated that maintaining a balanced diet plays a pivotal role in therapeutic contexts and in preventing chronic conditions related to aging, such as obesity, type 2 diabetes, cardiovascular diseases (CVD), and various forms of cancer [9,10,11,12,13].

Physical activity is another key factor in promoting health and preventing the onset of non-communicable diseases [14,15]. Indeed, consistent moderate physical exercise has been associated with multifaceted interventions yielding substantial benefits for health, with a number of investigations supporting a therapeutic potential for a range of chronic conditions, including both the primary and secondary prevention of CVD, obesity, type 2 diabetes, among others [16,17,18,19,20]. On the other hand, prolonged sedentary behavior has been shown to increase the risk of various chronic diseases associated with fat accumulation [21].

Combining changes in both diet and exercise habits yields more benefits than either alone concerning health and body adiposity [22]. In this context, the interplay between diet and physical activity across different populations and life stages has been analyzed [22,23]. This interaction effectively mitigates age-related cognitive decline [23]. Moreover, this synergy, along with psychosocial well-being, improves health and quality of life [24]. Furthermore, diet and physical activity interact with the gut microbiota, impacting health and exercise performance [25]. The associations between nutrition, physical activity, metabolic diseases, and environmental factors in the inflammation process have also been investigated, particularly the effect of the modification between them [26]. This approach has been implemented through personalized nutrition strategies that take into account these interconnected factors, aiming to address inflammation, oxidative stress, and immune responses more effectively [27,28]. In this context, old age has been characterized by a low-grade, chronic, inflammatory state, which has been termed “inflammaging” [28]. The complex metabolic interaction between caloric restriction and physical activity promotion with changes in body composition has also been studied [29], but the mechanism needs to be elucidated. Hence, pioneer research has explored the beneficial links between dietary patterns and physical exercise [30]. However, the precise interaction between diet and exercise in relation to human health and body composition remains unclear. This uncertainty is especially pronounced for BMI in large populations. Consequently, the development of combined precise lifestyle interventions is scarce, which makes it difficult to formulate tailored and personalized recommendations.

This cross-sectional study aimed to clarify the potential interaction between the Mediterranean diet and physical activity with BMI within an online cohort of adults. The purpose was to support precise nutritional and lifestyle interventions at both the individual and community levels. The participants were adults over 18 years old. This study specifically focused on transversally analyzing the possible effect of modification associated with interindividual variation in the levels of Mediterranean diet adherence and physical activity concerning adiposity.

## 2. Methods

### 2.1. Study Design and Sample

The NUTRiMDEA cohort was from a web-based survey performed between May 2020 and November 2020. This observational cross-sectional study included a total of 17,332 participants with the inclusion criteria of age being over 18 years old, having internet access, and understanding the Spanish language to complete the questionnaires. Data were collected through two different web-based methods (open survey and rewarded survey) to assemble information related to sociodemographic, lifestyle (Mediterranean diet, physical activity, and sleep patterns), HRQoL, and general health factors. Responders who completed the open survey, which was freely accessible online at https://nutrimdea2020.questionpro.com/, obtained a personalized report based on their habits and health. The questionnaires are described below. Further information was described elsewhere [31]. For the present analysis, only the open-survey participants were included (n = 11,883).

The survey questionnaire was presented to IMDEA-CEI (2019), which confirmed that participation in the questionnaire is a proof of acceptance to be involved in the NUTRiMDEA study with its own anonymized data. This research meets the applicable parts of the Declaration of Helsinki and Spanish guidelines/regulations (IVC/2020). More details of the protocol and ethical aspects have been described in previous publications [26,31]. At the end of the questionnaire, a sentence stated that by filling in the survey, they consent to the use of the provided data for scientific purposes as stated by the ethical committee (31 July 2019).

### 2.2. Participants

The inclusion criteria of this study were based on the acceptance of participation in the survey, being over 18 years of age, being a Spanish speaker, and consenting to the use of the data provided for scientific purposes. Interested participants were directed to the study website to complete the online questionnaire. All information collected is completely anonymous, and only the IP addresses were administered to avoid survey completion more than once from the same IP address. The completed questionnaires are finally encrypted and stored in local servers for exclusive access by the study researchers. A total of 17,332 participants were screened online from May 2020 to November 2020. Online ads such as internet websites, online newspapers, radio ads, and congresses were used to enroll volunteers. A total of 11,883 people completed the open survey, while the number of participants who encompassed the rewarded survey was 5449 that were excluded from this research.

### 2.3. Questionnaire and Measurements

The questionnaire encompassed sociodemographic data, self-reported anthropometric data, cardiometabolic disease prevalence, dietary habits, lifestyle, and physical and mental component scores of SF12 Health Survey (PCS/MCS, which ranged from 0—worse—to 100—better quality of life) [32,33]. All these variables were self-reported by the participants or calculated from the responses. 

Among sociodemographic data, age was analyzed within two age groups (<40 years and ≥40 years) and sex. The educational level was classified into two categories: high school or less (primary education; less, low, or intermediate secondary education; or higher secondary education) and more than high school (intermediate vocational education, higher vocational education, or university). 

Cardiometabolic disease prevalence was self-reported by the participants with the following question: “Have you been diagnosed or are you currently undergoing treatment for any of the following conditions?” The options were obesity, diabetes, high blood pressure, and dyslipidemia, and the possible answers were yes/no.

The BMI was calculated using data on self-declared weight and height, and individuals were stratified as having low or high BMI (<25.0 kg/m^2^—underweight and normal weight—or ≥25.0 kg/m^2^—overweight or obesity, respectively). 

For diet quality estimation, the Mediterranean adherence diet score was assessed by using the PREDIMED questionnaire, known as MEDAS-14 [34], as well as the number of meals a day, snacking habit, and glasses of water a day. In relation to lifestyle, nap habit, which is defined as a short period of sleep typically taken during daytime hours as an adjunct to the usual nocturnal sleep period, was categorized as yes/no, and sleep hours during weekdays and weekends were classified as <8 h or ≥8 h.

Physical activity was assessed using the International Physical Activity Questionnaire (IPAQ) using the Spanish version [35,36]. Activities were categorized as light, moderate, or intense based on the metabolic equivalent value (MET), which was converted to min/week (METs-min/w) and days/week (METs-d/w).

### 2.4. Statistical Analysis

Descriptive analyses concerning anthropometrics, biochemical, and lifestyle factors across sex, weight, age-specific, diet, and physical activity groups were performed. The normality of the variables was screened using the Shapiro–Wilk test. Descriptive statistics were given as median and interquartile ranks (IQR), and differences were assessed using the *t*-test or Mann–Whitney test for non-normal distribution. Categorical variables were reported as percentages and compared with the Chi-squared test. The median was used to stratify the population according to BMI and physical activity, while the cut-off point for the Mediterranean diet adherence questionnaire was 9, since adherence is considered low below this point [37]. Differences between groups and the presence of interactions were analyzed via factorial analysis of variance (ANOVA) for continuous variables (diet adherence × physical activity). Logistic or multinomial regression was used for categorical variables, if the variable had two or more categories, respectively. Subsequently, contrasts and linear hypothesis tests after estimation were performed. The multiple linear regression model was used to predict excessive body weight, with BMI as a proxy for obesity. The variables used in the regression models were sex, age, educational level, number of meals, snacking habit, water consumption, sleep hours, HRQoL (PCS and MCS), Mediterranean diet score, and physical activity, as well as potential interactions introducing the corresponding product terms to the models. The variance inflation factor (VIF) analysis for testing collinearity between independent variables was performed to ensure variable independence. To better illustrate the data concerning the ratio of physical activity/Mediterranean diet and BMI, the Y-axis was shown between 0 and 2500 units, and the X-axis between 10 and 55 kg/m^2^.

All *p*-values presented were two-tailed and considered statistically significant at *p*  <  0.05. Data were analyzed using STATA version 12.1 (StataCorp, College Station, TX, USA).

## 3. Results

The characteristics of the participants of the online open survey, separately analyzed by age, sex, and BMI, in the NUTRiMDEA study (n = 11,883) were mainly females (67.8%) and equal to or over 40 years old (65.7%), as reported in Table 1. The prevalence of obesity, diabetes, hypertension, and dyslipidemia were associated with the following factors: men, over 40 years old, and higher BMI (*p* < 0.001). The median BMI was higher for men than women (24.7 kg/m^2^ vs. 22.7 kg/m^2^) and for the participants over 40 years old than the younger ones (24.0 kg/m^2^ vs. 22.3 kg/m^2^). The dietary factors associated with high BMI were the Mediterranean diet score, number of meals, and snacking habits (*p* < 0.001). People with high BMI had less physical activity and more frequent naps (*p* < 0.001) than the lower BMI population. Significant differences have been observed in the hours of sleep between sex and age (*p* < 0.001), but not when classified by BMI (*p* = 0.1272). There were differences for PCS and MCS between BMI categories and age (*p* < 0.001); participants with low BMI declared better PCS, while the ones with high BMI had better MCS (Table 1).

When categorizing the population into four groups according to weight status (underweight, normal weight, overweight, and obese), statistically significant differences (*p* < 0.001) were found in the diet, with lower adherence to the Mediterranean diet in patients with excess weight. Likewise, it was observed that those with excess body weight performed more physical exercise than those with a lower BMI (*p* < 0.001).

The results after categorizing the sample by MEDAS and physical activity considering sex are reported in Table 2. The prevalence of hypertension and dyslipidemia was lower among women than among men, regardless of MEDAS score (*p* < 0.001), but the prevalence of diabetes showed this sex difference only for those with high MEDAS scores, showing an interaction between diabetes prevalence, Mediterranean diet, and sex (*p* < 0.01). No significant differences were observed in the reported prevalence of obesity between men and women concerning diet scores. However, this contrast became notable when comparing sexes within the low physical activity group, but men had higher BMI than women in all categories (Table 2). 

People with high MEDAS scores also had high physical activity. The median METs-day/w was 1.60 in men and 1.24 in women for the participants with low MEDAS scores and 2.25 in men and 1.68 in women for those with high MEDAS scores (*p* < 0.001). Nap habit on weekdays and weekends was more frequent among the men and the participants with high MEDAS scores (*p* < 0.001) but was not associated with the level of physical activity. Weekend napping habits (*p* = 0.0304) and PCS scores (*p* = 0.0465) showed interactions between sex, Mediterranean diet, and physical activity (Table 2). 

Differences in BMI between men and women are shown in Figure 1A with respect to their adherence to the Mediterranean diet and physical activity. In general, men had higher BMIs than women. As for the age classification, Figure 1B shows that those older than 40 years had a higher BMI. Both figures show that those who performed less physical activity and had a lower adherence to the Mediterranean diet showed a higher BMI. 

Therefore, the importance of both physical activity and the Mediterranean diet on adiposity is demonstrated. Increasing levels of physical activity predicted a decrease in BMI, as shown in Figure 2A, as did adherence to the Mediterranean diet, as reflected in Figure 2B. In addition, Figure 2C shows that an increase in physical activity levels induced an increase in the adherence to the Mediterranean diet, with its respective influence on adiposity. 

According to the increase in adiposity (BMI), the score of physical activity and Mediterranean diet decreases, which means that when physical activity is higher or the adherence to the Mediterranean diet is lower, the BMI is lower; when both decrease, the BMI is higher (Figure 3).

The predictive variables for an increase in BMI were age (older), snacking habit, and number of glasses of water. The predictive variables for decrease in BMI were sex (female), sleep hours, high PCS and MCS, high MEDAS and high physical activity and the interaction between MEDAS and physical activity. The interaction between Mediterranean diet and physical activity had a larger effect on BMI (Table 3). Data in Table 3 show that with increasing age, the number of meals and glasses of water per day, as well as the habit of snacking, the BMI was higher. However, an increase in sleeping hours, quality of life, adherence to the Mediterranean diet, and physical activity were associated with a lower BMI. Furthermore, being a woman may have a protective effect, since it decreased the BMI by 2 points.

BMI decreased as the level of physical activity increased, but the Mediterranean diet affected this relationship. The participants who had lower MEDAS scores, which represents worse quality of diet, exhibited higher BMI when the physical activity was low or non-existent compared to those with higher MEDAS scores. However, as physical activity levels increased, the BMI of those with lower diet quality were similar to that of participants with better dietary habits. Moreover, as the level of physical activity increased, BMI was equalized in both groups, regardless of adherence to the Mediterranean diet. Thus, patients with lower adherence to the Mediterranean diet decreased their BMI more rapidly with increased physical activity (Figure 4).

## 4. Discussion

The significant influence of lifestyle and milieu exposome, including the Mediterranean diet and exercise, on promoting healthy aging has been thoroughly examined across the life cycle [1,3,38,39], which can be boosted by using online approaches for personalized interventions at acceptable costs, suitability, and validity [40]. Evidence concerning the role of lifestyle factors highlights the importance of adopting healthy eating habits and active lifestyles early to promote healthy aging and long-term well-being and life quality [41]. Conversely, an unhealthy lifestyle characterized by physical inactivity, poor diet, and other factors starting from childhood can accelerate biological aging, highlighting the importance of addressing multiple aspects of lifestyle and personal history from an early age to promote healthier aging [42].

Interestingly, in this study, the ratio of physical activity/Mediterranean diet was lower when BMI was higher, suggesting that a decrease in physical activity leads to a higher BMI, whereas higher physical activity and/or lower adherence to the Mediterranean diet correlates with a lower BMI. This observation could imply that physical activity might have a more significant impact on BMI outcomes compared to adherence to the Mediterranean diet in this population. However, further investigation is needed to explore these results in greater detail, including an examination of the specific types of exercise undertaken (endurance or resistance exercise), since this study focused on total METs. Nevertheless, what was particularly noteworthy was the interaction between physical activity and the Mediterranean diet concerning BMI. Indeed, our findings hint that the detrimental impacts of a low adherence to the Mediterranean diet on BMI might be partly offset by higher levels of physical activity as a modification of the effect of unhealthy dietary patterns.

These findings are relevant to promoting physical activity, particularly in populations with low adherence to the Mediterranean diet or in those people for whom the Mediterranean diet intervention does not produce positive BMI results. Furthermore, it is advised to study the dietary patterns of the population, despite having an adequate BMI, since lower adherence to the Mediterranean diet could be masked by high levels of physical activity. This compensatory mechanism could hide the potential negative effects of a declining adherence to the Mediterranean diet, requiring a closer examination of both diet and activity levels to fully understand their combined impact on general health. The metabolically unhealthy normal-weight phenotype is now recognized, which may pose the same or even a greater risk of cardiometabolic disease compared to overweight individuals [43]. Therefore, in normal weight populations, it is highly recommended to perform metabolic and lifestyle screening. This approach may help prevent future risks of cardiovascular disease and mortality.

Another relevant finding of this research was the differences found between the physical activity/Mediterranean diet ratio concerning BMI, suggesting that exercise may have a more substantial influence than Mediterranean diet in maintaining adequate levels of weight, and the interplay between the level of physical exercise and dietary habits on BMI, indicating that a higher level of physical activity might mitigate the detrimental effects of a poor-quality diet on BMI outcomes. Indeed, the impact of the Mediterranean diet and physical activity on adiposity depends on both factors, with differential outcomes associated with body composition (adiposity).

On the other hand, HRQL has been demonstrated as a crucial instrument in assessing the perceived health of the population as a valuable health marker surrogate [44,45]. In this context, differences emerged based on age. These dissimilarities may stem from better mental health and greater psycho-social adaptability among older adults, particularly amid the COVID-19 pandemic, as suggested by previous research [46,47,48]. While PCS exhibited no differences between men and women, MCS was greater in males, as previously reported in the literature [49,50]. A statistical interaction was observed between the Mediterranean diet, physical activity, and sex concerning PCS. These findings emphasize the significance of considering interactions among various sociodemographic and lifestyle factors [50,51]. Concerning BMI, PCS was greater with a lower BMI consistent with other authors [5], whereas MCS increased with higher BMI. It is important to note that the BMI-HRQL relationship follows an inverse U-shaped pattern, peaking at a BMI of 22 kg/m^2^ for women and plateauing at BMI values of 22–30 kg/m^2^ for men. Older men (aged 50 years and above) with a BMI of 29 kg/m^2^ reported an average five-point higher score compared to those with a BMI of 20 kg/m^2^. This epidemiological pattern is more pronounced in older individuals and varies across ethnicities [51]. The tendency for napping was heightened in the group of men, aged ≥40 years and among individuals with the highest BMI. These findings may differ from those of another study that found a positive correlation between napping and a reduced risk of obesity [52].

Age and sex can interact with different health-related factors, such as physical activity and diet, and the magnitude of these can influence BMI [21,53,54]. In this regard, we noted variations in health characteristics and lifestyles within our study population, stratified by age and sex. Our findings were compared with the National Health Survey of Spain (Encuesta Nacional de Salud de España, 2017), agreeing that women had a worse mood but a healthier lifestyle regarding diet or smoking and had lower rates of overweight [55]. Also, the older participants exhibited higher BMI values, consistent with existing literature [56]. Furthermore, these results were accentuated based on adherence to the Mediterranean diet and physical activity levels within our online population, with higher BMI levels when adherence to the Mediterranean diet or levels of physical exercise were lower, regardless of sex or age (both men and women and under and over 40 years old). This feature suggests a uniform dynamic across all groups in this regard.

As expected, the linear regression analyses revealed that various sociodemographic and lifestyle factors were associated with BMI in our population, including age, sex, education level, snacking habit, sleep patterns, HRQoL, Mediterranean diet, and physical activity, as previously described in the literature [57]. The previous results are also corroborated here by demonstrating that physical activity shows a stronger association with the BMI results compared to the Mediterranean diet.

Moreover, our data suggested that both a poor-quality diet and low physical activity were independently linked to a higher BMI, aligning with the existing literature [58]. Also, a high adherence to the Mediterranean diet correlated with increased physical activity levels, which is beneficial. 

The interaction demonstrated in this study between physical activity and the Mediterranean diet on BMI, alongside the lifestyle differences related to age and sex, implies that strategies promoting physical activity and the Mediterranean diet could benefit from tailored considerations based on age and sex. Also, this online methodology of data collection, coupled with the use of bioinformatic tools to introduce new health strategies, may facilitate reaching broader populations and implementing more personalized and precision interventions [30,59]. This approach could involve addressing age- or sex-related disparities in exercise preferences, motivation, and perceived obstacles, among other factors. Indeed, a mediation study may help customize health messages to be more individual-specific and may enhance the effectiveness in promoting healthy lifestyles and weight management, thus contributing to healthier aging.

However, the current study has some limitations, as the causality of the observed associations cannot be determined as part of an observational study. Online enrollment could involve a technological bias in some population groups. Although there is already available literature validating this type of online methods [60], it has been described both in this research and in other investigations that a percentage of people tend to slightly underestimate their own weight and increase their height [61], but the validity of the results has been found to be acceptable. The use of an online method for data collection can be criticized for not being fully representative of the whole population, as it excludes those who are computer illiterate and those without internet access [62]. However, online data collection has been used in other surveys such as the SUN cohort [63], Food4Me [60], Nurses’ Health study [64], Health Professionals follow-up study [65,66], and PROM study [59], which supports the validity of the results obtained with a population recruited online. The influence of the COVID-19 pandemic coinciding with data collection cannot be discarded, so there may be biases in some data.

Although online studies offer advantages such as easy recruitment and broad geographic reach, their applicability to other populations must be carefully evaluated. Considerations such as access to technology, sample diversity and representativeness, and sociocultural and economic differences are critical in determining the relevance and effectiveness of research in different settings. Additional adaptations and validations may be necessary to ensure the applicability of the results to diverse populations. The strengths of our study included our large sample size; however, it is important to recognize the possibility of both type I and type II errors.

## 5. Conclusions

A reciprocal interplay between the Mediterranean diet, exercise, and BMI has been established, alongside the modifying influence of dietary patterns and physical activity on BMI within an adult online population (18 to >70 years old). These findings corroborate the significance of tailoring lifestyle recommendations for healthy aging within the population, considering potential interactions among different factors, in pursuit of precision nutrition and medicine, where the Mediterranean diet and exercise may differently influence each other depending on body weight.

## Figures and Tables

**Figure 1 nutrients-16-01777-f001:**
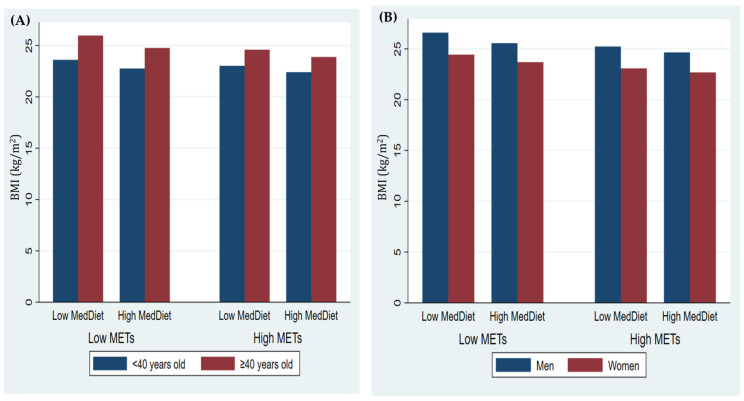
Mean BMI score based on adherence to the Mediterranean diet and physical activity according to (**A**) sex and (**B**) age.

**Figure 2 nutrients-16-01777-f002:**
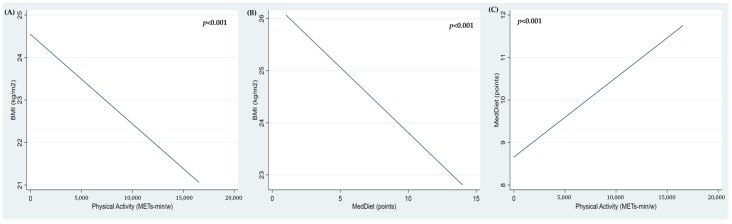
Linear prediction models of the effect of physical activity (**A**) and MedDiet (**B**) on BMI. Mutual influence of physical activity and adherence to the Mediterranean diet (**C**).

**Figure 3 nutrients-16-01777-f003:**
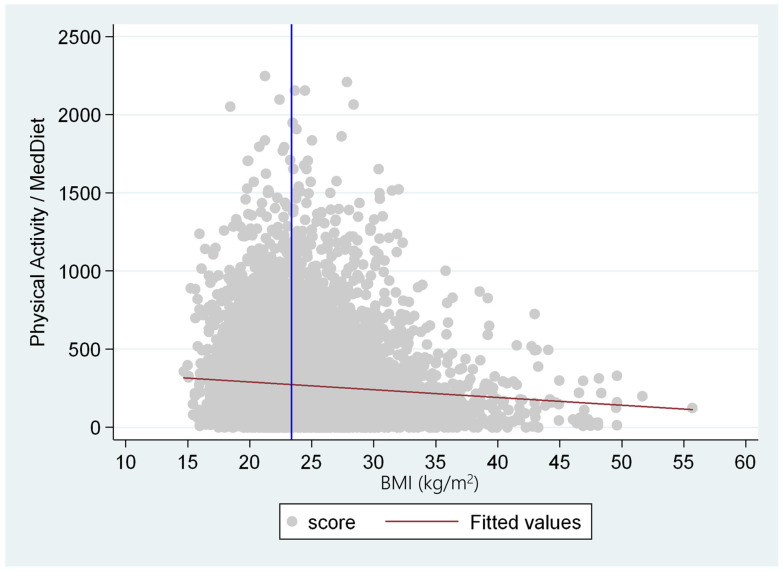
Effect of the physical activity/MedDiet score on the BMI. The blue line represents the BMI median and the red line the slope of the association with the physical activity/MedDiet score.

**Figure 4 nutrients-16-01777-f004:**
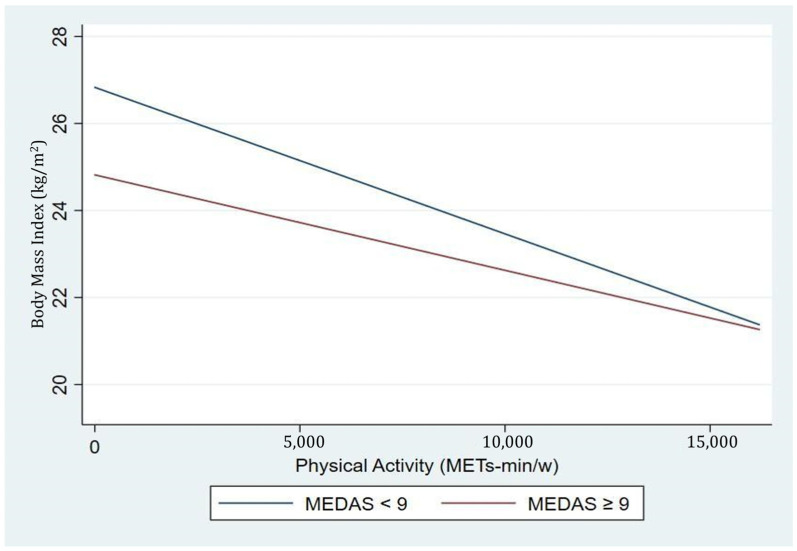
Effect of the changes in BMI and physical activity by Mediterranean diet score in the NUTRiMDEA population.

**Table 1 nutrients-16-01777-t001:** General characteristics, anthropometric markers, dietary habits, and lifestyle factors of the populations stratified by age, sex, and body mass index.

	Age			Sex			BMI		
	<40 Years Old	≥40 Years Old	*p*-Value	Men	Women	*p*-Value	Low	High	*p*-Value
**n**	**4079**	**7804**		**3826**	**8057**		**5953**	**5930**	
General characteristics									
Sex (%)			**<0.001**			**-**			**<0.001**
Men	30.89	69.11		100	0		32.33	67.67	
Women	35.96	64.04		0	100		58.53	41.47	
Age (%)			**-**			**<0.001**			**<0.001**
<40 years old	100	0		28.98	71.02		63.89	36.11	
≥40 years old	0	100		33.88	66.12		42.89	57.11	
Education (%)			**<0.001**			**<0.001**			**0.001**
High school or less	5.56	94.44		51.39	48.61		31.94	68.06	
More than high school	35.43	64.57		31.38	68.62		50.97	49.03	
Obesity (%)	30.4	5.55	**<0.001**	5.15	4.47	0.1009	0.44	13.48	**<0.001**
Diabetes (%)	0.66	2.95	**<0.001**	3.14	1.70	**<0.001**	1.35	3.85	**<0.001**
Hypertension (%)	1.37	11.75	**<0.001**	12.89	5.96	**<0.001**	4.12	16.61	**<0.001**
Dyslipidemia (%)	6.13	20.52	**<0.001**	19.34	13.79	**<0.001**	12.55	21.85	**<0.001**
Anthropometric measurement									
Weight (kg)	63.0 (55.0;73.0)	68.0 (59.0;78.0)	**<0.001**	77.0 (70.0;85.0)	61.0 (55.0;68.0)	**<0.001**	60.0 (55.0;67.0)	80.0 (73.0;89.0)	**<0.001**
BMI (kg/m^2^)	22.27 (20.42;24.61)	24.01 (21.80;26.73)	**<0.001**	24.69 (22.84;27.13)	22.67 (20.66;25.28)	**<0.001**	21.99 (20.43;23.44)	27.53 (26.04;29.78)	**<0.001**
Dietary habits									
MEDAS (points)	9 (8;10)	9 (8;11)	**<0.001**	9 (8;10)	9 (8;11)	0.8575	9 (8;11)	9 (8;10)	**<0.001**
Number of meals (%)			**<0.001**			**<0.001**			**<0.001**
1 or 2 meals	31.61	68.39		50.91	49.09		43.52	56.48	
3 meals	31.11	68.89		35.26	64.74		48.76	51.24	
4 meals	37.68	62.32		27.81	72.19		52.97	47.03	
5 or more meals	37.48	62.52		25.42	74.58		50.80	49.20	
Snacking habits (%)	50.36	42.75	**<0.001**	40.98	47.44	**<0.001**	44.00	48.26	**<0.001**
Water consumption (%)			**<0.001**			0.5109			0.1547
1–4 glasses/day	28.86	71.14		31.83	68.17		50.50	49.50	
≥5 glasses/day	37.59	62.41		32.41	67.59		49.86	50.14	
Lifestyle factors									
Physical activity (METs-min/w)	2381.95 (2054.44)	2460.57 (2166.25)	0.3483	2910.20 (2459.52)	2207.22 (1911.13)	**<0.001**	2580.00 (2173.26)	2130.55 (2001.09)	**<0.001**
Nap habit (%)	21.92	32.19	**<0.001**	36.49	24.95	**<0.001**	26.04	34.05	**<0.001**
Sleep weekday (%)			**<0.001**			**<0.001**			0.1272
≤8 h	33.89	66.11		32.45	67.55		49.97	50.03	
>8 h	48.99	51.01		23.63	76.37		54.47	45.53	
PCS (points)	56.06 (53.41;58.34)	55.29 (51.32;56.98)	**<0.001**	55.49 (53.39;57.05)	55.49 (51.63;57.44)	0.5261	56.06 (53.35;57.78)	54.99 (51.02;56.71)	**<0.001**
MCS (points)	44.82 (31.53;50.62)	48.69 (39.60;52.51)	**<0.001**	49.61 (40.92;53.12)	46.62 (34.37;51.55)	**<0.001**	46.98 (35.72;51.56)	48.28 (36.90;52.51)	**<0.001**

Variables are shown as median (IQR) according to its distribution. Continuous variables were compared using *t*-test. Categorical variables are shown as proportion (%) and compared using Chi-squared test. Significant values are in bold font. BMI, body mass index; MEDAS, Mediterranean diet adherence screener; METs, metabolic equivalents, PCS, physical component summary; MCS, mental component summary.

**Table 2 nutrients-16-01777-t002:** General characteristics, anthropometric markers, dietary habits, and lifestyle factors of the populations classified by adherence to Mediterranean diet and physical activity according to sex.

	MEDAS					Physical Activity					
	<9 Points		≥9 Points		*p*-Value **	Low		High		*p*-Value **	*p*-Value ***
	♂	♀	♂	♀		♂	♀	♂	♀		
**n**	**1355**	**2724**	**2471**	**5333**		**1564**	**4372**	**2262**	**3685**		
Age (%)					0.305					0.724	0.792
<40 years old	30.83	69.17 *	27.75	72.25 *		23.08	76.92 *	29.91	70.09 *		
≥40 years old	34.80	65.20	33.46	66.54		25.76	74.24	35.15	64.85		
Education (%)					0.374					0.272	0.792
High school or less	59.26	40.74 *	46.67	53.33 *		66.67	33.33 *	49.21	50.79		
More than high school	32.35	67.65	30.88	69.12		24.35	75.65	32.48	67.52		
Obesity (%)	6.94	6.13	4.17	3.62	0.365	7.35	5.63 *	3.63	3.09	0.053	0.967
Diabetes (%)	2.80	2.42	3.32	1.33 *	**0.003**	3.64	1.88 *	2.79	1.49 *	0.859	0.244
Hypertension (%)	11.88	5.91 *	13.44	5.98 *	0.371	15.60	6.68 *	11.01	5.10 *	0.397	0.139
Dyslipidemia (%)	18.45	13.44 *	19.83	13.97 *	0.690	21.93	15.42 *	17.55	11.86 *	0.804	0.580
Anthropometric measurement											
Weight (kg)	80.96 (15.25)	64.12 (12.71) *	77.49 (12.10)	62.36 (10.85) *	**<0.001**	81.06 (15.04)	64.34 (12.47) *	77.10 (11.88)	61.31 (10.09) *	0.055	0.227
BMI (kg/m^2^)	25.96 (4.33)	23.95 (4.54) *	24.94 (3.48)	23.16 (3.81) *	0.182	26.03 (4.38)	23.98 (4.45) *	24.80 (3.31)	22.77 (3.51) *	0.869	0.702
Dietary habits											
MEDAS (points)	6.90 (1.28)	7.08 (1.14) *	10.29 (1.16)	10.17 (1.08) *	**<0.001**	8.56 (2.03)	8.79 (1.84) *	9.46 (1.93)	9.52 (1.73)	**0.030**	0.570
Number of meals (%)					0.563					0.563	0.196
1 or 2 meals	51.07	48.93*	50.75	49.25 *		42.17	57.83 *	53.30	46.70 *		
3 meals	36.75	63.25	34.39	65.61		28.15	71.85	36.48	63.52		
4 meals	27.94	72.06	27.75	72.25		18.43	81.57	29.09	70.91		
5 or more meals	20.53	79.47	27.22	72.78		13.06	86.94	26.87	73.13		
Snacking habits (%)	46.86	54.22 *	37.76	43.97 *	0.653	44.63	50.59 *	38.46	43.69 *	0.773	0.753
Water consumption (%)					0.075					0.109	0.475
1–4 glasses/day	31.62	68.38	31.98	68.02		24.75	75.25	33.39	66.61		
≥5 glasses/day	34.47	65.53	31.50	68.50		24.91	75.09	33.33	66.67		
Lifestyle factors											
Physical activity (METs-min/w)	2301.43 (2232.04)	1782.37 (1778.86) *	3244.02 (2514.35)	2424.27 (1939.86) *	**<0.001**	981.98 (561.29)	935.59 (550.84) *	4243.41 (2380.30)	3716.34 (1852.14) *	**<0.001**	0.143
Nap habit weekday (%)	32.47	24.01 *	38.69	25.43 *	**0.030**	34.65	24.11 *	37.75	25.94 *	0.671	0.534
Nap habit weekend (%)	32.47	24.01 *	38.69	25.43 *	**0.030**	34.65	24.11 *	37.75	25.94 *	0.671	**0.030**
Sleep weekday (%)					0.661					0.124	0.557
≤8 h	33.58	66.42 *	31.87	68.13 *		25.13	74.87	33.58	66.42 *		
>8 h	23.24	76.76	23.90	76.10		18.99	81.01	25.00	75.00		
PCS (points)	55.2 (51.5;57.0)	55.2 (50.9;57.5) *	55.6 (53.1;57.1)	55.6 (52.1;57.4) *	0.895	55.0 (51.1;56.7)	55.0 (50.5;57.0)	55.9 (54.4;57.3)	56.1 (53.6;57.8)	0.147	**0.047**
MCS (points)	48.2 (37.6;52.5)	44.3 (30.9;50.5) *	50.2 (43.1;53.5)	47.5 (36.5;51.8) *	0.776	48.7 (37.9;52.4)	45.6 (32.6;51.0) *	50.2 (43.1;53.5)	47.5 (36.6;51.9) *	0.947	0.559

Variables are shown as median (IQR) according to its distribution. Continuous variables were compared using *t*-test. Categorical variables are shown as proportion (%) and compared using Chi-squared test. Significant values are in bold font. * *p*-value of the difference between men and women; ** *p*-value of the difference between the four groups; *** *p*-value of the interaction between Mediterranean diet score, sex, and physical activity. MEDAS, Mediterranean diet adherence screener; BMI, body mass index; METs, metabolic equivalents, PCS, physical component summary; MCS, mental component summary; ♂, men; ♀, women.

**Table 3 nutrients-16-01777-t003:** Univariate linear regression model of body mass index (BMI) as dependent variable.

	Body Mass Index (BMI)		
	*β*	*p*-Value	R^2^
		**<0.001**	0.1765
Age (years)	1.048	**<0.001**	
Sex (M/W)	−2.001	**<0.001**	
Educational level	0.117	0.105	
Number of meals	0.096	**0.020**	
Snacking (yes/no)	0.306	**<0.001**	
Water (glasses)	0.288	**<0.001**	
Sleep weeks (hours)	−0.182	**<0.001**	
PCS (points)	−0.132	**<0.001**	
MCS (points)	−0.016	**<0.001**	
MedDiet (points)	−0.286	**<0.001**	
Physical Activity (METs-d/w)	−0.508	**<0.001**	
MedDiet#METs	0.024	**0.047**	

*β* represents changes in outcome (BMI) for the increasing number of units of the predictive variables. Bold numbers indicate statistical significance (*p* < 0.05). PCS, physical component summary; MCS, mental component summary; METs, metabolic equivalent task.

## Data Availability

The datasets presented in this article are not readily available because of data protection regulations. Data can be made available in a de-identified manner to researchers upon reasonable request (to the extent allowed by the registry’s data protection agreement). Requests to access these datasets should be directed to the NUTRiMDEA study primary investigator, J. Alfredo Martínez (jalfredo.martinez@imdea.org).

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
