# Peer review of "Reciprocal and Differential Influences of Mediterranean Diet and Physical Activity on Adiposity in a Cohort of Young and Older than 40 Years Adults"

_nutrients, 2024, doi:10.3390/nu16111777_

Round 1

Reviewer 1 Report

Comments and Suggestions for Authors

Please see pdf.

Comments on the Quality of English Language

Please see pdf.

Author Response

Review 1 Report Form

Open Review

Quality of English Language

( ) I am not qualified to assess the quality of English in this paper
( ) English very difficult to understand/incomprehensible
( ) Extensive editing of English language required
( ) Moderate editing of English language required
(x) Minor editing of English language required
( ) English language fine. No issues detected

Yes

Can be improved

Must be improved

Not applicable

Does the introduction provide sufficient background and include all relevant references?

( )

( )

(x)

( )

Are all the cited references relevant to the research?

(x)

( )

( )

( )

Is the research design appropriate?

( )

(x)

( )

( )

Are the methods adequately described?

( )

(x)

( )

( )

Are the results clearly presented?

( )

( )

(x)

( )

Are the conclusions supported by the results?

( )

( )

(x)

( )

Comments and Suggestions for Authors

 The present study aimed to assess the influence of physical and diet on BMI in a large cohort (11,883 participants). In my opinion, this is an important topic that requires further attention to inform the development of personalized recommendations. This is also acknowledged by the authors. However, the nature of the study (cross-sectional data) and the way physical activity data are presented preclude from drawing firm conclusions or inform personalized recommendations despite the large sample size. A revised analysis will be helpful to better understand the impact of physical activity on BMI. My comments are as follows:

  1. Abstract: The authors state that “The combined effect of diet and physical activity had a greater impact on BMI than physical activity alone”. What about diet? Please add “or diet alone”.

We appreciate this request. The abstract has been drafted as follows:

  1. Abstract (Page 1)

The combined effect of diet and physical activity had a greater impact on BMI than physical activity or Mediterranean diet adherence alone, with lower BMI as physical activity levels increased (β: -0.5), but differently influenced by diet (β: -0.28).

  1. Abstract: The type of diet “Mediterranean” is mentioned once in the abstract. If you solely focused on Mediterranean diet, consider specifying throughout the abstract and manuscript.

We thank for this suggestion. The tittle has been reworded as follows:

  1. Title Page (Page 1)

Reciprocal and differential influences of Mediterranean Diet and physical activity on adiposity in a cohort of young and older than 40 years adults

The term “Mediterranean” has been also added throughout the whole manuscript as appropriate.

  1. Abstract: It is stated that participants with lower adherence to Mediterranean diet displayed higher BMI when physical activity was low, but as physical activity rose, their BMI aligned with heathier dietary habits. This sentence gives the impression of a longitudinal study (e.g., as physical activity rose). However, this was a cross-sectional study. Please revise to enhance clarity.

We kindly thank you for this comment. The sentence has been rewritten to improve clarity. In addition, it has been clarified that this is a cross-sectional study.

  1. Abstract (Page 1)

Data from 11,883 NUTRiMDEA cohort participants were analyzed in this cross-sectional study, categorizing individuals by age, sex, and BMI using linear regression models to assess interactions between lifestyle factors and adiposity.

Participants with lower adherence to Mediterranean diet displayed superior BMI when physical activity was low, but when activity levels were higher, their BMI aligned with those with healthier dietary habits.

  1. Introduction (first paragraph): I would encourage the authors to avoid using vague terms such as “metabolic well-being”.

Thank you for this comment. The term “metabolic well-being” and other terms have been removed from the manuscript to make it easier to read and understand, and replaced by “healthy aging” as follows:

  1. Introduction (Page 1)

However, greater longevity does not guarantee a higher quality of life and healthy aging [1].

  1. Introduction: The introduction is somewhat long (8 paragraphs) and needs to be revised to improve clarity. There is a lot of information on various topics, such as aging, lifestyle, physical activity, diet, inflammation, and online data collection. From the reader’s perspective, it is a little difficult to follow after the fourth paragraph given the diversity of the information provided. I do not think that paragraph #6 is necessary (i.e., online data collection) in the introduction. Many studies have employed this strategy. You can simply describe how data were collected in the methods section. Also, I would suggest that you remove the term “online” from the title of the manuscript. The flow requires improvement as well. For example, on paragraph 5, the authors state that “combining changes in both diet and exercise habits yield more benefits than either alone concerning health and body adiposity”. Then, on paragraph 7, the information is repeated. Moreover, many sentences include many words which is somewhat distracting.

Examples: “Nonetheless, the specific interplay between diet and exercise in relation to human health and body composition remains uncertain.….personalized recommendations”.

The last paragraph consists of a very long 5-line sentence.

The study aim, as it is currently stated is very complex and must be simplified.

Overall, I would highly encourage the authors to provide the most relevant information needed before transitioning to the methods section.

We appreciate this suggestion.

The term online has been removed from the title. In the introduction section, some paragraphs and sentences have been reduced, combined, and rewritten to make them easier to read. The introduction currently consists of six paragraphs.

  1. Introduction (Page 2)

Hence, pioneer research has explored the beneficial links between dietary patterns and physical exercise [30]. However, the precise interaction between diet and exercise in relation to human health and body composition remains unclear. This uncertainty is especially pronounced regarding BMI in large populations. Consequently, the development of combined precise lifestyle interventions is scarce, which makes it difficult to formulate tailored and personalized recommendations.

This cross-sectional study aimed to clarify the potential interaction between the Mediterranean diet and physical activity with BMI within an online cohort of adults. The purpose was to support precision nutritional and lifestyle interventions at both individual and community levels. Participants who ranged in age from 18 to over 70 years old. The study specifically focused on analyzing the possible effect modification associated with the variation in individual levels of mediterranean dietary adherence and physical activity concerning adiposity.

Also, the paragraph containing the information on “online data collection” has been removed from the manuscript.

  1. Introduction: The authors have cited studies that examined the link between diet and physical activity as it relates to BMI. However, they mention that “the specific interplay between diet and physical activity in relation to human health and body composition remains uncertain, particularly concerning BMI”. Can you please clarify what is currently unknown about the relationship between diet and physical activity and their impact on BMI?

Thanks for your observation. We consider that it is necessary to clarify the objective assessment and quantification of the modifying effect. Also, to expound the different impact of physical activity and Mediterranean diet depending on baseline BMI.

  1. Questionnaires and Measurements: My following point relates to lengthy sentences. I would encourage the authors to avoid redundant information. For example, “sex consisted of two categories (men and women)”. Please note that if you refer to “sex”, this should be followed by “males” and “females”. I would remove entirely this sentence.

Thanks for the suggestion. The sentence has been removed to avoid repeated material. The content of this section has been reviewed and redundant information has been removed.

  1. How did you decide to dichotomize BMI based on 25.0 kg/m2? Given the large sample size, you could have created at least 4 categories to better describe the following BMI categories: underweight, normal weight, overweight, and obese. Additionally, this gives the impression that BMI was treated as a dichotomous variable. However, you used multiple linear regression to look at changes in BMI units. Please revise accordingly to clearly describe your approach.

We would like to express our gratitude for this comment. In the regression model, the BMI variable was used as a continuous linear variable. However, in the tables the population has been classified into two groups according to the BMI cutoff because, according to the World Health Organization (WHO), 25kg/m2 or more is considered overweight, which is associated with multiple comorbidities. Furthermore, only 3.81% of the population is in the underweight group, so the 4 groups are very different (63.34% normal weight and 24.91% overweight and 7.94% obese). Nevertheless, new analyses have been made with the BMI categorized into 4 categories (underweight, normal weight, overweight, and obese).

  1. Results (Page 4)

When categorizing population into 4 groups according to weight status (underweight, normal weight, overweight and obese), statistically significant differences (p<0.001) were found in the diet, with lower adherence to the Mediterranean diet in patients with excess weight. Likewise, it was observed that those with excess body weight performed more physical exercise than those with a lower BMI (p<0.001).

  1. For physical activity, can you further describe the use of the median? Also, you have reported METs-d/w in tables but not METs-min/week which is clearly more relevant. I would strongly suggest including METs-min/week in the analysis instead of METs-d/wk. Additionally, if possible, it would be useful to indicate how your physical activity categories relate to the physical activity recommendations. What does “high” or “low” METs mean in Table 2 and figure 1? Was this based on MET-d/wk or MET-min/wk?

Thank you for this comments and questions.

For physical activity, the median was used as the cutoff point in order to categorize the population into two comparable groups.

As suggested it has been modified to METs-min/w in the tables, as in Figure 4, but METs-d/w has been maintained in the logistic regression since otherwise the β coefficient was too small to be understood.

The physical activity levels of the NUTRiMDEA population have been compared with the WHO guidelines, being recommended habits between 150-300 minutes per week of moderate physical activity. A moderate physical activity is classified with an expenditure of between 3-6 METs, therefore, the WHO recommendations translated into METs would be between 675-1350 (calculating with 4.5 METs which is the average between 3 and 6). (For 3 METs: 450-900 METs-min/w; for 6 METs: 900-1800 METs-min/w). On the other hand, the PA guidelines recommend between 500 and 1000 METs per week.

'High' and 'Low' METs refers to the variable dichotomized into 2 groups, stratified by the median, with the 'High' group having the highest values from the median and 'Low' having the lowest values from the median. In both Table 2 and Figure 1 the data are shown in METs-min/w.

  1. In the statistical analysis section, you stated that physical activity was included in the multiple regression model. Does table 3 list the coefficients of the multivariable or the univariate analysis for each predictor? Please use MET-m/wk instead of MET-d/wk in the multivariable analysis.

We appreciate this comment. In the logistic regression, METs-d/w is used because otherwise the coefficient β was too small to be applicable. We are dealing with a univariate analysis.

  1. How did you compare between-group differences in tables 1 and 2? Please add to statistical analysis.

Thanks for the suggestion. This has been explained in the first paragraph of statistical analysis in the material and methods section, adding a clarifying paragraph. A sentence has also been added at the foot of the table.

  1. Methods (Page 4 – Statistical analysis)

Differences between groups and the presence of interactions were analyzed by factorial analysis of variance (ANOVA) for continuous variables (diet adherence x physical activity). Logistic or multinomial regression was used for categorical variables, if the variable had two or more categories, respectively. Subsequently, contrasts and linear hypothesis tests after estimation were performed.

  1. The results need to be better explained. For example, you state that the dietary factors associated with high BMI were Mediterranean Diet Score, number of meals, and snacking habits. Can you please provide direction for these variables (e.g., lower or higher Mediterranean Diet Score and what does it mean for adherence)?

We appreciate this comment. A paragraph and appropriate amendments have been included in the results section where the results of table 3 are explained in more detail.

  1. Results (Page 10)

Data in Table 3 show that with increasing age, the number of meals and glasses of water per day, as well as the habit of snacking, the BMI was higher. However, an increase in sleeping hours, quality of life, adherence to the Mediterranean diet and physical activity was associated with a lower BMI. Furthermore, being a woman may have a protective effect, since it decreased the BMI by 2 points.

  1. For the interaction between diet and physical activity in table 3, did you use MET-d/wk or MET-min/wk?

The logistic regression uses METs-d/w because otherwise the coefficient β was too modest to be comprehensible.

  1. The discussion diverts from the study objective. For example, the interaction of diet and physical activity with BMI is briefly discussed towards the end of this section. The discussion must be streamlined in line with the study objectives to improve clarity.

We appreciate the reviewer's insightful suggestion. Therefore, we have rearranged the order of the interaction-related paragraphs at the beginning of the discussion. Additionally, we have included more information on this point to align with the study objectives and improve clarity.

  1. Discussion (Page 11, line 13, 2nd paragraph)

These findings are relevant to promoting greater physical activity, particularly in populations with low adherence to the Mediterranean diet or in those people for whom the Mediterranean diet intervention does not produce positive results on BMI. Furthermore, it would be advisable to study the dietary patterns of the population, despite having an adequate BMI, since lower adherence to the Mediterranean diet could be masked by high levels of physical activity. This compensatory mechanism could hide the potential negative effects of a declining Mediterranean dietary pattern, requiring closer examination of both diet and activity levels to fully understand their combined impact on health. The metabolically unhealthy normal weight phenotype is now recognized, which may pose the same or even greater risk of cardiometabolic disease compared to overweight individuals [43]. Therefore, in normal weight populations, it is highly recommended to perform metabolic and lifestyle screening. This approach may help prevent future risk of cardiovascular disease and mortality.

We have also reduced (new paragraph 4 and 7) and clarified (6) other paragraphs in the discussion.

  1. Please include study limitations in the discussion. In the original version, no study limitations are discussed.

In response to your comment, a paragraph has been added at the end of the discussion section with the limitations of the study.

  1. Discussion (Page 13)

However, the current study has some limitations, as the causality of the observed associations cannot be determined due to be an observational study with a potentially non-representative sample as well as the online enrollment could involve a technological bias in some population groups. Although there is already available literature validating this type of online methods [60], it has been described both in this research and in other investigations that a percentage of people tend to slightly underestimate their own weight and increase their height [61] but the validity of the results has been found to be acceptable. The use of an online method for data collection can be criticized for not being fully representative of the whole population, as it excludes computer illiterate and those without internet access [62]. However, online data collection has been used in other surveys such as the SUN cohort [63], Food4Me [60], Nurses’ Health study [64] and Health Professionals follow-up study [65, 66] or PROM study [59], which supports the validity of the results obtained with a population recruited online. The influence of the COVID-19 pandemic coinciding with data collection cannot be discarded, so there may biases some data.

Although online studies offer advantages such as easy recruitment and broad geographic reach, their applicability to other populations must be carefully evaluated. Considerations such as access to technology, sample diversity and representativeness, and sociocultural and economic differences are critical in determining the relevance and effectiveness of research in different settings. Additional adaptations and validations may be necessary to ensure the applicability of the results to diverse populations. Strengths of our study included our large study size; however, it is important to recognize the possibility of both type I and type II errors.

Reviewer 2 Report

Comments and Suggestions for Authors

This manuscript represents an observational study correlating diet, lifestyle factors, body mass index, and health-related quality of life. The authors utilize the NUTRiMDEA cohort unknown to assess associations across the lifespan and compare individuals less than age 40 to those greater than age 40. Gerontologists with would comparisons for subjects over age 65 of more interest.  Can the authors compare young individuals less than age 40 to older individuals over age 65?  Can the authors compare 3 age categories, less than age 40, age 40-64, age 65 and above?

In the methods section the authors should identify the origin of the study population, recruitment strategy and which individuals were excluded using a PRISMA recruitment diagram.

The abstract should include some data regarding the major findings.

A limitations section should follow the discussion and include information related to the population studied, applicability to other populations, a disclaimer that observational relationships do not imply causality, and recommendations for future studies.

Author Response

Review 2 Report Form

Open Review

Quality of English Language

( ) I am not qualified to assess the quality of English in this paper
( ) English very difficult to understand/incomprehensible
( ) Extensive editing of English language required
( ) Moderate editing of English language required
( ) Minor editing of English language required
(x) English language fine. No issues detected

Yes

Can be improved

Must be improved

Not applicable

Does the introduction provide sufficient background and include all relevant references?

(x)

( )

( )

( )

Are all the cited references relevant to the research?

(x)

( )

( )

( )

Is the research design appropriate?

( )

(x)

( )

( )

Are the methods adequately described?

( )

(x)

( )

( )

Are the results clearly presented?

( )

(x)

( )

( )

Are the conclusions supported by the results?

( )

(x)

( )

( )

Comments and Suggestions for Authors

This manuscript represents an observational study correlating diet, lifestyle factors, body mass index, and health-related quality of life. The authors utilize the NUTRiMDEA cohort unknown to assess associations across the lifespan and compare individuals less than age 40 to those greater than age 40. Gerontologists with would comparisons for subjects over age 65 of more interest.  Can the authors compare young individuals less than age 40 to older individuals over age 65?  Can the authors compare 3 age categories, less than age 40, age 40-64, age 65 and above?

We thank for this suggestion. It certainly would be interesting to make 3 categories as you mention, however, unfortunately our database does not have a cutoff point at age 65 as our ranges are: 18-25 years old, 25-40 years old, 40-55 years old, 55-70 years old and over 70 years old.

In the methods section the authors should identify the origin of the study population, recruitment strategy and which individuals were excluded using a PRISMA recruitment diagram.

We would like to thank for this comment. A paragraph has been added with information about the participants; however, since there are no exclusion criteria other than that the survey was a rewarded survey, we do not believe it is necessary to include a PRISMA diagram.

  1. Methods (Page 3)

Participants

The inclusion criteria of the study were based on the acceptance of participation in the survey, being over 18 years of age, Spanish speaker and consenting to the use of the data provided for scientific purposes. Interested participants were directed to the study website to complete the online questionnaire. All information collected is completely anonymous, only the IP addresses were administered to avoid survey completion more than once from the same IP address. The completed questionnaires are finally encrypted and stored in local servers for exclusive access by the study researchers. A total of 17,332 participants were screened online from May 2020 to November 2020. Online ads such as internet websites, online newspapers, radio ads and congresses were used to enroll volunteers. A total of 11,883 people completed the open survey, while the number of participants who encompassed the rewarded survey was 5,449 that were excluded from this research.

The abstract should include some data regarding the major findings.

Thank you for this comment. The abstract has been slightly modified, but due to limitations in the number of words (max. 200) it has not been possible to extend the results further.

  1. Abstract

The combined effect of diet and physical activity had a greater impact on BMI than physical activity or Mediterranean diet adherence alone, with lower BMI as physical activity levels increased (β: -0.5), but differently influenced by diet (β: -0.28).

A limitations section should follow the discussion and include information related to the population studied, applicability to other populations, a disclaimer that observational relationships do not imply causality, and recommendations for future studies.

We kindly thank you for this comment. In response to your point, a paragraph has been added at the end of the discussion section with the limitations of the study, a disclaimer that observational relationships do not imply causality, its applicability to other populations and recommendations for the future.

  1. Discussion (Page 13)

However, the current study has some limitations, as the causality of the observed associations cannot be determined due to be an observational study with a potentially non-representative sample as well as the online enrollment could involve a technological bias in some population groups. Although there is already available literature validating this type of online methods [60], it has been described both in this research and in other investigations that a percentage of people tend to slightly underestimate their own weight and increase their height [61] but the validity of the results has been found to be acceptable. The use of an online method for data collection can be criticized for not being fully representative of the whole population, as it excludes computer illiterate and those without internet access [62]. However, online data collection has been used in other surveys such as the SUN cohort [63], Food4Me [60], Nurses’ Health study [64] and Health Professionals follow-up study [65, 66] or PROM study [59], which supports the validity of the results obtained with a population recruited online. The influence of the COVID-19 pandemic coinciding with data collection cannot be discarded, so there may biases some data.

Although online studies offer advantages such as easy recruitment and broad geographic reach, their applicability to other populations must be carefully evaluated. Considerations such as access to technology, sample diversity and representativeness, and sociocultural and economic differences are critical in determining the relevance and effectiveness of research in different settings. Additional adaptations and validations may be necessary to ensure the applicability of the results to diverse populations. Strengths of our study included our large study size; however, it is important to recognize the possibility of both type I and type II errors.

Furthermore, a paragraph on ethical aspects has been added to the methods section:

  1. Methods (Page 3)

The survey questionnaire was presented to IMDEA-CEI (2019), which confirmed that participation in the questionnaire is a proof of acceptance to be involved in the NUTRiMDEA study with own anonymized data. In any case, this research accomplishes the applicable Helsinki declaration and Spanish guidelines/regulations (IVC/2020). In addition, more details of the protocol and ethical aspects have been described in previous publications [26, 31]. At the end of the questionnaire a sentence stated that by filling the survey, they consent to the use of provided data for scientific purposes as called on by the Ethical Committee (31/07/2019).

Round 2

Reviewer 1 Report

Comments and Suggestions for Authors

Thank you for reviewing and addressing my comments. I only have a few suggestions for the revised version prior to publication:

1. Abstract: Please locate the following excerpt: "with lower BMI as physical activity levels increased (β: -0.5), but differently influenced by diet (β: -0.28)..."

The text in bold "but differently influenced" gives the impression that Mediterranean diet was associated with a higher BMI. Please revise to improve clarity. 

2. The last paragraph of the introduction needs improvements to enhance clarity. The sentence "Participants who ranged in age from 18 to over 70 years old" seems random. Also, in the last sentence, it is important to specify whether you're referring to intra- or inter-individual variability. The former implies a longitudinal analysis whereas the latter a cross-sectional study, in line with your study design. 

3. Table 3: Add the word "Univariate" before "Linear regression" in the table's title. 

Comments on the Quality of English Language

N/A

Author Response

Thank you for reviewing and addressing my comments. I only have a few suggestions for the revised version prior to publication:

  1. Abstract: Please locate the following excerpt: "with lower BMI as physical activity levels increased (β: -0.5), but differently influencedby diet (β: -0.28)..."

The text in bold "but differently influenced" gives the impression that Mediterranean diet was associated with a higher BMI. Please revise to improve clarity. 

Thank you for the comment. The abstract has been modified as follows:

  1. Abstract (Page 1)

The combined effect of diet and physical activity had greater impact on BMI than physical activity or Mediterranean diet adherence alone, with lower BMI as physical activity levels increased (β: -0.5), as with adherence to Mediterranean diet decrease, where a modification effect between then was identified (β: -0.28).

  1. The last paragraph of the introduction needs improvements to enhance clarity. The sentence "Participants who ranged in age from 18 to over 70 years old" seems random. Also, in the last sentence, it is important to specify whether you're referring to intra- or inter-individual variability. The former implies a longitudinal analysis whereas the latter a cross-sectional study, in line with your study design. 

We appreciate this suggestion. The sentence of the participants’ age has been rewritten to make it clear.

  1. Introduction (Page 2)

Participants were adults over 18 years old.

The term “interindividual” has been added to the last sentence of the introduction and it also clarified that this is a cross-sectional study as follows:

  1. Introduction (Page 2)

The study specifically focused on transversally analyzing the possible effect modification associated with the interindividual variation in levels of mediterranean dietary adherence and physical activity concerning adiposity.

  1. Table 3: Add the word "Univariate" before "Linear regression" in the table's title. 

The term “univariate” has been incorporated to the title of table 3.

  1. Table 3 (Page 10)

Table 3. Univariate linear regression model of Body Mass Index (BMI) as dependent variable.

Reviewer 2 Report

Comments and Suggestions for Authors

The authors are in substantial compliance with reviewer comments.

Minor typos noted - suggestions for correction in attached file.

Author Response

The authors are in substantial compliance with reviewer comments.

Minor typos noted - suggestions for correction in attached file.

We appreciate for this comment. All required typing recommendations have been made.

The last sentence of the second paragraph of the material and methods section cannot be deleted because it is a recurring request from the editor.
